# Investigating the Benefit of Combined Androgen Modulation and Hypofractionation in Prostate Cancer

**DOI:** 10.3390/ijms21228447

**Published:** 2020-11-10

**Authors:** Alice Zamagni, Michele Zanoni, Michela Cortesi, Chiara Arienti, Sara Pignatta, Antonella Naldini, Anna Sarnelli, Antonino Romeo, Anna Tesei

**Affiliations:** 1Biosciences Laboratory, Istituto Scientifico Romagnolo per lo Studio e la Cura dei Tumori (IRST) IRCCS, 47014 Meldola, Italy; michele.zanoni@irst.emr.it (M.Z.); michela.cortesi@irst.emr.it (M.C.); chiara.arienti@irst.emr.it (C.A.); sara.pignatta@irst.emr.it (S.P.); 2Cellular and Molecular Physiology Unit, Department of Molecular and Developmental Medicine, University of Siena, 53100 Siena, Italy; antonella.naldini@unisi.it; 3Medical Physics Unit, Istituto Scientifico Romagnolo per lo Studio e la Cura dei Tumori (IRST) IRCCS, 47014 Meldola, Italy; anna.sarnelli@irst.emr.it; 4Radiotherapy Unit, Istituto Scientifico Romagnolo per lo Studio e la Cura dei Tumori (IRST) IRCCS, 47014 Meldola, Italy; antonino.romeo@irst.emr.it

**Keywords:** prostate cancer, extreme hypofractionated radiotherapy, hypoxia, senescence-associated secretory phenotype (SASP), DNA repair

## Abstract

Hypofractionation is currently considered a valid alternative to conventional radiotherapy for the treatment of patients with organ-confined prostate cancer. Recent data have demonstrated that extreme hypofractionation, which involves the use of a high radiation dose per delivered fraction and concomitant reduction of sessions, is a safe and effective treatment, even though its radiobiological rationale is still lacking. The present work aims to investigate the biological basis sustaining this approach and to evaluate the potential of a hypofractionated regimen in combination with androgen deprivation therapy, one of the major standards of care for prostate cancer. Findings show that androgen receptor (AR) modulation, by use of androgens and antiandrogens, has a significant impact on cell survival, especially in hypoxic conditions (4% O_2_). Subsequent experiments have revealed that AR activity as a transcription factor is involved in the onset of malignant senescence-associated secretory phenotype (SASP) and activation of DNA repair cascade. In particular, we found that AR stimulation in hypoxic conditions promotes the enhanced transcription of *ATM* gene, the cornerstone kinase of the DNA damage repair genes. Together, these data provide new potential insights to justify the use of androgen deprivation therapy, in particular with second-generation anti-androgens such as enzalutamide, in combination with radiotherapy.

## 1. Introduction

Prostate cancer (PCa) is the second most commonly diagnosed cancer in males, accounting for 7.1% of cancer worldwide [1]. Although substantial efforts have been made to identify risk factors it is likely that a complex interplay of genetic and lifestyle factors must be taken into account [2,3]. With regard to PCa management, radiotherapy is one of the most important treatment options and the use of a hypofractionated regimen is currently becoming an area of interest, because of its intrinsic characteristics. Hypofractionated radiotherapy (HFRT), consisting of daily larger-sized fractions and a consequently shorter treatment duration [4], positively impacts on treatment costs and patients’ quality of life, two of the most important factors to consider when making treatment decisions. The recently published ASTRO, ASCO and AUA guidelines for localized PCa define two types of HFRT: moderate-HFRT (m-HFRT), which consists of 2.4–3.4 Gy/daily fractions, and extreme-HFRT (e-HFRT), which consists of ≥5 Gy/daily fractions [5]. In particular, e-HFRT is potentially comparable to the high-dose rate of a brachytherapy approach, with the advantage of being less invasive and more economically convenient [6]. Several Phase I and II trials focusing on e-HFRT have recently focused on the management of low and intermediate PCa, with excellent results in terms of short-term toxicity and long-term biochemical control [7,8]. Of note, single doses > 8 Gy are even more advantageous because of their ability to trigger direct endothelial cell death and depletion of nutrients [9,10]. Important Phase III clinical trials are still ongoing or have given recent results, comparing the use of moderate and extreme hypofractionation. In particular, the international, randomized, open-label PACE-B non-inferiority trial reported the absence of increased toxicity (gastrointestinal or genitourinary acute toxicity) in the e-HFRT group [11]. Late toxicity and biochemical control results will be released within 2–3 years.

A combination approach of androgen deprivation therapy (ADT) and radiotherapy currently represents the standard of care, in particular for high-risk patients [12,13,14,15]. Numerous clinical trials have reported an improved disease-free and overall survival in patients dealing with high or intermediate risk disease undergoing combined treatment, compared to radiotherapy alone. The radiobiological principles underlying such a therapeutic effect are still not clearly understood [16]. 

Moreover, a hypoxic microenvironment is a central key component of PCa [17,18,19]. Hypoxia has a detrimental effect on the efficacy of treatment and is correlated with resistance to radiotherapy because of the markedly reduced effects of radiation in the absence of oxygen [20,21]. The impact of hypofractionation should also be considered from a radiobiological point of view. Tumor shrinkage, resulting from the eradication of cells, is likely to be observed with the use of conventional radiotherapy (2 Gy/fraction), where the majority of sensitive cells are killed during irradiation and stop consuming oxygen [22]. When using HFRT, the reoxygenation phenomenon cannot occur through tumor shrinkage because of the shorter time of treatment duration [23,24]. A better understanding of spatial and temporal dynamics of hypoxia in tissue is needed to define the outcomes of radiation treatment. Within this context, progress in imaging of tumor hypoxia has been achieved through the use of positron emission tomography (PET). Furthermore, improved detection and imaging of hypoxia in vivo has led to the development of different strategies for escalating radiation doses to hypoxic regions in tumors [25,26].

In the present work we aimed to investigate the radiobiological mechanisms sustaining e-HFRT, because of its still undefined therapeutical advantages with respect to conventional radiotherapy. Accordingly, we evaluated the benefit of e-HFRT in combination with ADT, as data from the literature do not directly address this issue.

## 2. Results

### 2.1. Cell Line Characterization 

LNCaP, PC3 and LNCaP R-bic cells were characterized at baseline condition, both in normoxia (21% O_2_) and after 6 months of continuous exposure to hypoxic conditions (4% O_2_) (Appendix A). In agreement with the literature data, hormone-sensitive LNCaP expressed high amounts of androgen receptor (AR), with active transcriptional activity, as shown by high levels of the classic AR-related target gene, prostate specific antigen (PSA). AR and PSA expression was similar in both oxygen conditions. For this reason, LNCaP was considered as our model of biochemical naive-PCa. In LNCaP R-bic, a cell line obtained from LNCaP by continuous exposure to 20 μM R-bicalutamide, AR maintained a homogeneous expression between normoxia and hypoxia. Interestingly, this cell line continued to express PSA protein under normoxic and hypoxic conditions and can thus be defined as a castration-resistant model of PCa (CRPC). Finally, the PC3 cell line showed no expression of AR and PSA, proving an excellent model of AR-negative PCa. As expected, HIF-1α expression was absent in normoxic cells and present in hypoxic counterparts.

Lastly, PCa cells maintained in hypoxic conditions had a notably higher proliferation rate, with respect to their normoxic counterparts, except for PC3 (Appendix A).

### 2.2. AR Expression and Modulation Impacts on Cell Proliferation

We first evaluated the impact of single and combination treatments on cell proliferation in the month following 10 Gy dose to gain a better understanding of the survival scenario. Figure 1 underlines the differences between hypoxic (grey, A, C, E) and normoxic (white, B, D, F) conditions.

In particular, in irradiated LNCaP grown in hypoxic conditions (Figure 1A), androgen stimulation by R1881 plays a consistent protective role against radiotherapy, with surviving fraction between untreated control cells and 10 Gy alone cells (0.80 vs. 0.47, respectively, at week 4). Similarly, ADT, with either R-bicalutamide or MDV3100 (the latter proved to be a more potent molecule), caused a synergistic effect with radiotherapy, resulting in poor survival rates (0.21 at week 4 in “MDV3100+10Gy”-treated cells). By contrast, this trend was not seen under normoxic condition (Figure 1B), as the radiation doses, coupled with the presence of oxygen, exert the leading role and keep cell survival very low, but fairly homogeneous, making it impossible to discriminate between the different androgen modulations.

Figure 1C,D show survival rates of LNCaP R-bic cell line, in hypoxic and normoxic conditions, respectively. At week 4, hypoxia gave homogeneous results for combined treatment, whereas in normoxic conditions, the presence of oxygen contributed to striking cell death, with the highest disease control reached in the “MDV3100+10Gy” condition, as frequently observed in clinical practice. Of note, in hypoxic condition the survival rates of LNCaP R-bic were higher than those of LNCaP, confirming the more aggressive nature of this CRPC model.

The same experimental plan was applied to the AR-negative PC3 to validate the putative role of AR in response to radiotherapy. As expected, AR modulators did not affect cell survival when used alone, either in hypoxic (Figure 1E) or normoxic conditions (Figure 1F). In combined treatments, comparable results were obtained with respect to 10 Gy alone, in both oxygen conditions, suggesting that AR may be involved in cell recovery and response to a hypofractionated radiotherapy regimen.

### 2.3. Role of AR in DNA Damage Repair after e-HFRT

To investigate the putative role of AR in cell recovery after e-HFRT, we used the comet assay to quantify the amount and the level of DNA damage resulting from each condition receiving 10 Gy fraction. Figure 2 shows the results of the comet assay performed on cell lines under both oxygen conditions. Representative images of all conditions are reported on the left, while the bar charts indicate the mean of the overall damage, expressed as the percentage of DNA in the comet tail. 

In line with previous considerations, hypoxic LNCaP cells (Figure 3A) receiving the “R1881+10Gy” combination showed a significantly lower total damage compared to the same cells receiving 10 Gy alone. As reported in the bar chart, the “R1881+10Gy” condition led to an intermediate percentage of DNA in the comet tail (1.26 ± 0.11) with respect to control (0.63 ± 0.11) and 10 Gy fraction alone (5.28 ± 0.83). Conversely, the normoxic LNCaP cells counterpart (Figure 2B) showed higher total DNA damage values, as indicated in the bar chart. Furthermore, a higher proportion of class 3 and 4 DNA damage (representative of mostly total and total DNA fragmentation, respectively) is described in Appendix A. 

With regard to the other cell lines, i.e., LNCaP R-bic (Figure 2C,D, respectively) and PC3 (Figure 2E,F, respectively) under both hypoxic and normoxic conditions, some general considerations can be made by looking at survival rates. First, “MDV3100+10Gy” conditions of normoxic and hypoxic LNCaP R-bic have similar trends with respect to 10 Gy alone, highlighting the poor impact in terms of DNA damage that a second line drug coupled with radiotherapy had on CRPC cell lines. Moreover, normoxic LNCaP R-bic undergoing “MDV3100+10Gy” showed the highest DNA damage, considering also that comet class 4 alone (see Appendix A) represents alone over 75% of all the classes, in line with the very low survival rates reported in Figure 1D. Normoxic LNCaP R-bic cells fared the worst, reflecting what happens in CRPC patients in clinical practice. 

To give weight to the hypothesis of a promising role of AR in cell recovery after e-HFRT, we performed a comet assay on the AR-negative, androgen-insensitive PC3, observing no significant differences between the combined treatment and 10 Gy alone under either hypoxic or normoxic conditions. 

Once we had established the influence of AR modulation on cell recovery and DNA damage repair, we assessed the influence of AR on cell cycle and checkpoint control to define a cell status profile. We also wanted to verify if the radiosensitivity of the “ADT+10Gy” combinations might be due, in part, to the synchronization of cells in the more radiosensitive phases of the cell cycle, i.e., mainly S and G2 phase (Appendix A). No significant differences between the various cell phases were detected in any of the cell lines, leading us to conclude that the difference in DNA damage cannot be ascribed to cell cycle.

Lastly, AR expression and modulation had a high impact on the cells’ ability to repair DNA damage and, as a consequence, to escape cell killing after radiotherapy.

### 2.4. Impact of Androgen Receptor Signaling on DDR Gene Expression Following High-Radiation Dose Exposure in Hypoxic Milieu

Given the promising findings that emerged from hypoxic LNCaP, we decided to focus all further experiments on this cell line. 

We decided to use real-time PCR to perform an in-depth analysis of AR-dependent gene expression after irradiation, focusing in particular on DNA damage repair (DDR) genes, HIF-1α target genes and inflammation-related genes. Classic AR target genes are shown in the far left block of the heatmap in Figure 3A. 

We observed that AR modulation in irradiated cells activated a finely regulated expression program, with interesting results in terms of DNA repair and inflammation pathways. In particular, *ataxia telangiectasia-mutated* gene (*ATM*) expression and fluctuation was perfectly in line with our hypothesis. ATM is the cornerstone kinase that rapidly localizes to sites of DNA damage in response to DNA double-strand breaks (DSBs). Figure 3B shows the main tightfitting genes whose modulation was dependent on AR stimulation or inhibition. In particular, hypoxic LNCaP showed significantly increased levels of *ATM* when treated with “R1881+10Gy” rather than 10 Gy alone. *PSA* expression was also tested as a positive control to confirm that AR was still functioning and transcriptionally active, despite irradiation. Accordingly, a higher expression of PSA was detected in “R1881+10Gy”-treated cells than those exposed to R1881 alone. 

We considered *ATM* modulation an important result, as ATM serine/threonine kinase is a master regulator of the DNA DSB repair pathway. We also noticed that chemokine receptor *CXCR4* gene expression, implicated in PCa progression and the metastatization process, followed a similar AR-dependent modulation. 

As previously underlined, DNA damage and DSBs are major consequences of radiotherapy. The inability to resolve DNA DSBs leads cells to arrest proliferation and eventually undergo apoptosis. This, together with the promising data of the *ATM* modulation obtained by real-time PCR, prompted us to perform a Western blot analysis of DDR proteins to evaluate their function and involvement (Figure 3C). A striking upregulation of ATM was detected also at protein level, confirming and strenghtening the real-time PCR results. Intriguingly, we also observed a consistent upregulation of RAD51 in the “R1881+10Gy” condition, compared to 10 Gy alone. Of note, cells treated with “MDV3100+10Gy”showed a marked reduction in RAD51 expression. RAD51 plays a major role in the homologous recombination (HR) pathway of DNA repair. The same trend was found for p-Chk2, whose phosphorylation is directly performed by ATM, confirming that the entire pathway was strongly influenced by AR modulation.

### 2.5. Enhancement of Senescence-Associated Secretory Phenotype (SASP) by Androgen Stimulation in Irradiated Hypoxic LNCaP Cells

The observation of cell recovery in the first week after irradiation leads us to notice a condition-specific behavior. As expected, “ADT+10Gy” treatments were characterized by increased levels of apoptosis (Figure 4A) and modulated expression of pro-apoptotic and anti-apoptotic markers (Figure 4B). These results were obtained 144 h hours after irradiation and the apoptotic data were confirmed by a long-term viability assay (Figure 1A).

On the other hand, we also noticed an unusual arrest of proliferation in irradiated LNCaP cells stimulated with R1881 (Figure 4A).

Given the fact that cells receiving “R1881+10Gy” also appeared larger and flatter than cells undergoing the other treatments, we hypothesized that this might be due to androgens mediating a pathway of cellular senescence after the 10 Gy dose. We thus measured senescence-associated (SA) β-gal activity 96 h after 10 Gy in all combination treatments in order to see whether cellular senescence was induced (Figure 4C). Although all irradiated cells showed an averagely enhanced induction of senescence (with respect to androgen modulation alone, Appendix A), interestingly, “R1881+10Gy” treatment resulted in a significantly higher percentage of blue cells than 10 Gy alone, thus supporting our hypothesis. 

These data were also corroborated by the results of gene expression analysis (Figure 4D), which revealed an upregulated expression of key genes involved in the occurrence of the SASP (senescence-associated secretory phenotype). An androgen-mediated cellular-senescence is established 72 h after 10 Gy dose, upregulating IL-6, TNF-α, C/EPB and CXCL-10. Taken together, these findings indicate that “R1881+10Gy” cells, benefiting from SASP, had more time to repair DNA damage (also partially AR-mediated) before starting to grow again, showing better proliferative rates with respect to 10 Gy alone.

In light of the previously discussed ATM upregulation in “R1881+10Gy” cells, we finally wanted to verify if ATM inhibition had consequences on the SASP onset, besides DNA repair. For this reason, we exposed cells to a potent and selective ATM inhibitor, i.e., KU-55933. We measured SA β-gal activity 96 h hours after 10 Gy and noticed that senescence was decreased, compared to “R1881+10Gy” (Figure 4E), demonstrating that the AR-dependent ATM upregulation exerts an important role in the establishment of the SASP phenotype, representing an interesting strategy by a clinical point of view. Accordingly, gene expression analysis confirmed the inhibition of this phenotype, as reported in Figure 4F. DNA damage analysis was also performed, confirming an increase of total damage and high grade comet classes, as shown in Appendix A.

## 3. Discussion

HFRT is nowadays considered a safe and effective alternative to conventional radiotherapy for the management of PCa patients [8,27]. Notably, e-HFRT is emerging as a promising therapeutic approach that involves the use of a high radiation dose per delivered fraction. Lastly, in response to the COVID-19 outbreak, emerging data encourage the use of e-HFRT, or at least consider its feasibility, in order to limit the access of cancer patients in hospitals and minimize the potential spread of the pandemic [28,29,30]. Although many Phase I and II clinical trials on e-HFRT have demonstrated excellent disease control in addition to good short-term efficacy and safety, this radiotherapeutic regimen is still not considered a standard of care for PCa patients because of the lack of long-term safety assessment. Several Phase III trials are currently ongoing to address this issue [16]. Furthermore, a validated mechanistic understanding of the biological basis for the therapeutic effect of e-HFRT is still lacking and the aim of the present work was to contribute to filling this gap. Our findings are graphically presented in Figure 5.

In particular, we investigated the benefits of a combined ADT and e-HFRT treatment (10 Gy dose) on three PCa cell lines, representative of different stages of disease and cultured under different oxygen conditions (4% O_2_, hypoxia or 21% O_2_, normoxia) to obtain a clearer picture of the PCa scenario, which is characterized by large hypoxic areas [31,32,33]. We found that AR modulation, in combination with e-HFRT, regulated a fine cellular and transcriptional cascade, with visible modulations in DNA damage repair, senescence, cell survival and cell proliferation. A long-term viability assay, performed to mimic the clinical follow-up of the combined use of ADT and e-HFRT, revealed that hypoxic LNCaP cells recovered better from radiation therapy if the AR was stimulated by synthetic hormone R1881. Curiously, the same cells seemed to undergo a better resolution of DNA damage and enhanced activation of major DDR genes, including ATM and RAD51. An opposite trend was seen for the combined use of enzalutamide (formerly MDV3100) and 10 Gy, i.e., enhanced apoptosis with respect to radiation dose alone and poorer recovery from DNA damage. This outcome is of particular interest, according to the long-time debate concerning the combination of conventional radiotherapy and ADT. This strategy has been adopted as a standard option for men dealing with high-risk PCa, who could in this way benefit from both biochemical control and prolonged disease-free survival [34]. 

Interestingly, our data were supported by the presence of two simultaneous conditions: a transcriptionally active AR and a hypoxic microenvironment. Given the sensitivity of the response, we focused our research on the presence of hypoxic conditions, in particular HIF-1, because no changes were seen in AR protein expression between hypoxia and normoxia cells at baseline characterization. Moreover, the coexistence of these two important oncogenic axes was not surprising, as evidence has emerged of a possible synergism between AR and HIF-1α, both considered finely regulated transcription factors. For example, hypoxia increases AR transactivation in the presence of dihydrotestosterone (DHT) and the expression of PSA mRNA is higher in the presence of both conditions than in the presence of androgen alone or hypoxia alone. In fact, in addition to several androgen-responsive elements (AREs), the PSA promoter region contains a functional hypoxia-responsive element (HRE) to which HIF-1 binds in hypoxic conditions [35]. Furthermore, AR is reported to increase HIF-1-mediated gene expression in LNCaP cells, in the presence of DHT and to sustain the transcription of major HIF-1 targets such as GLUT-1 and VEGF, the former involved in glucose uptake and the latter in vascular endothelial growth [36,37]. 

We next focused on the establishment of SASP, significantly increased in the “R1881+10Gy” treatment. SASP, a paracrine-based signaling used by cancer cells to escape anticancer therapies, is responsible for the expression of a spectrum of soluble factors leading to inflammation, immunosuppression and driving tumor progression [38,39,40]. The activation of AR is known to be involved in establishing senescence via the recruitment and upregulation of p21 and a contemporary decrease of phospho-Rb and p63 [41]. Other studies have demonstrated the AR-dependent modulation of the p16-Rb-E2F1 axis and Src/PI3K pathways [42]. We found that ATM activation is required for establishing senescence in hypoxic LNCaP cells, as the pharmacological inhibition of ATM activity prevent the occurrence of SASP and enhance DNA damage, as we expected. This last result is particularly intriguing, as the development of new therapeutical strategies targeting DNA DSBs are crucial in target therapy innovation [43,44,45], in particular when radiotherapy is given. Two first-in-class oral selective ATM inhibitors are currently in Phase I trials. AZD0156 and its enhanced version, AZD1390, are currently in use both in monotherapy or in combination (NCT02588105; NCT03423628), encouraged by striking preclinical results [46,47].

However, to the best of our knowledge, no data are available when a combination of ADT and e-HFRT is used. In such a complex scenario, we observed after irradiation the highest cell survival and induction of SASP in hypoxic prostate cancer cells with hormone-stimulated AR, supporting the hypothesis that SASP cells may mediate tumor growth by secreting factors that significantly promote tumorigenicity in neighboring malignant cells. 

## 4. Materials and Methods

### 4.1. Cell Lines and Population Doubling Time

The human prostate cancer cell lines LNCaP and PC3, both derived from metastatic sites (lymph node and bone, respectively) were purchased from the American Type Culture Collection (ATCC). LNCaP R-bic is a bicalutamide-resistant cell line derived from LNCaP and isolated in our laboratory [48]. Cells were grown in RPMI medium supplemented with 10% fetal bovine serum and glutamine (2 mM). In addition, LNCaP-Rbic was maintained in continuous exposure to 20 µM (R)-bicalutamide. Cells were periodically checked for mycoplasma contamination by MycoAlert™ Mycoplasma Detection Kit (Lonza, Milan, Italy).

To determine cell proliferation doubling time (PDT), 8 × 10^4^ cells/well were plated into 6-well plates and counted after 24, 48, 72, 96 and 120 h. The exponential growth equation y = z·e^kx^, obtained by graphing the mean of the number of cells/well (*y*-axis) and time point (*x*-axis), was used to obtain PDT with the formula ln2/k.

The Trypan blue exclusion test was used to evaluate the percentage of viable cells, which always exceeded 98% for the duration of the experiment.

### 4.2. In Vitro Radiation System

Flasks were inserted into a custom-built plexiglass phantom (40 × 40 × 8 cm), which was irradiated using a 6-MV photon beam delivered by an Elekta Synergy linear accelerator (Elekta Oncology Systems, Stockholm, Sweden). The delivered dose was calculated using the Philips Pinnacle 3 radiation therapy planning system (Philips Healthcare, DA Best, The Netherlands) customized with the geometric and dosimetric characteristics of an Elekta Synergy linear accelerator [49].

### 4.3. Treatments

#### 4.3.1. Oxygen Conditions

Cells were maintained at 37 °C at normoxic conditions (21% O_2_, atmospheric oxygen ~160 mmHg) or hypoxic conditions (4% O_2_, ~30 mmHg). Hypoxic cell lines had been cultured for at least 6 months under hypoxia before performing the experiments.

#### 4.3.2. Drugs 

(R)-bicalutamide, the active enantiomer of Casodex, was purchased from Vinci Biochem and resuspended in acetone (Merck Millipore Science S.r.l., Milan, Italy). MDV3100 was kindly provided by Dr. Greta Varchi (CNR-ISOF Institute of Bologna, Italy) and resuspended in DMSO (Merck Millipore, Milan, Italy). Synthetic androgen R1881 was purchased by Glentham Life Sciences and resuspended in DMSO (Merck Millipore, Milan, Italy). KU-55933 was purchased from Tocris Bioscience (Bristol, United Kingdom) and resuspended in DMSO. All chemicals were dissolved to a final concentration of 10 mM, divided into aliquots and stored at −20 °C. Drugs were freshly diluted in cell culture medium immediately before use. MDV3100, KU-55933 and (R)-bicalutamide were tested at 20 µM while R1881 was tested at physiological conditions (10 nM). The final concentration of solvents never exceeded 1% and did not interfere with cell behavior. 

#### 4.3.3. Combination Treatments

Cells were seeded in flasks and made quiescent using 5% charcoal-stripped fetal bovine serum (Sigma Aldrich) and medium lacking phenol red, for 24 h. Thereafter, one of the above described antiandrogens (MDV3100 or (R)-bicalutamide) or R1881 was added to the medium for 24 h, after which the cells were irradiated. For the experimental set involving KU-55933, it was added to the medium together with R1881, 24 h hours prior to irradiation to allow ATM inhibition before 10 Gy.

#### 4.3.4. Radiation 

Cells were irradiated using the linear acceleration Elekta Synergy Platform system. The delivery dose used was 10 Gy, mimicking an extreme hypofractionation fraction. Then, 72 h after irradiation, cells were trypsinized and processed for the experiments described below.

### 4.4. In Vitro Cytotoxicity Assay

Cell viability was assessed by Sulforhodamine B (SRB) assay according to the method proposed by Skehan et al [50]. Cells were seeded into a 96-well plate at a density of 2 × 10^3^ cells/well and the effects of the different combined treatments were evaluated 144 h after the 10 Gy dose. Experiments were run in octuplicate and each experiment was repeated twice. The optical density (OD) of treated and control cells was determined at a wavelength of 490 nm using a colorimetric plate reader.

### 4.5. Soft Agar Assay

Cells (1 × 10^3^ per well) were mixed with 0.4% Seaplaque agar in growth medium supplemented with antibiotics and plated on top of a solidified layer of 1% Agar Noble in HBSS medium supplemented with antibiotics in a 24-well plate. Six series of samples were prepared for each treatment dose. Freshly prepared growth medium, containing or not containing AR modulators, was added once a week. Colonies with more than 50 cells were quantified weekly under inverted microscope (Olympus IX51 microscope, Olympus Corporation, Tokyo, Japan) by two independent observers.

### 4.6. Cell Cycle

Cells were fixed in ethanol (70%) and stained in a solution containing 100 μg/mL of propidium iodide (PI) (Sigma Aldrich), 10000 units/mL of RNase (Sigma Aldrich) and 0.01% NP40 (Sigma Aldrich) overnight at 4 °C in the dark. After 24 h, samples were analyzed by flow cytometry. Data were elaborated using Modfit (DNA Modelling System) software (Verity Software House Inc., version 4.1.7, Topsham, ME) and expressed as fractions of cells in the different cell cycle phases. Flow cytometric analysis was performed using FACSCanto flow cytometer (BD, San Diego, CA, USA). Data acquisition and analysis were performed using FACSDiva software (BD). Samples were run in triplicate (10,000 events for each replica). Data express the fractions of cells in the different cycle phases and report the mean value of two experiments with standard deviations.

### 4.7. Alkaline Comet Assay

The assay was performed according to the manufacturer’s protocol (Comet assay, Trevigen, Gaithersburg, MD, USA). Briefly, 3000 cells were suspended in LM Agarose (at 37 °C) at a ratio of 1:10 (*v*/*v*) and 35 μL was immediately transferred onto the comet slide and left at 4 °C for gelling time. The slides were then immersed for 1 h at 4 °C in a pre-chilled lysis solution, washed in the dark for 1 h at 4 °C in alkaline unwinding solution, then electrophoresed for 30 min at 21 V. Slides were immersed twice in dH_2_O for 5 min each, then dipped in 70% ethanol and stained with 20 μL of diluted SYBR® Green Master Mix (BioRad, Milan, Italy). At least one hundred comets spanning from category 0 to category 4 were evaluated by EVOS Cell Imaging Systems 10x (Thermo Fisher Scientific, Waltham, MA, USA). DNA damage was quantified by computing the displacement between the genetic material contained in the nucleus, typically representing the “comet head”, and the genetic material in the surrounding part, considered as the “comet tail”. Percentage of DNA in tail for different categories of comets was expressed as previously described by O. Garcia et al [51]. In order to obtain reproducible and reliable data, we previously developed an easy-to-use tool named CometAnalyser [52].

### 4.8. RNA Extraction and Quantitative Polymerase Chain Reaction (RT-qPCR)

Total RNA was extracted from cell pellets using TRIzol^®^ Reagent, according to the manufacturer’s instructions (Invitrogen, Thermo Fisher Scientific, Waltham, MA, US). RNA was quantified using the Nanodrop® ND-1000 spectrophotometer system and stored at −80 °C. Reverse transcription reactions were performed in 20 µL of nuclease free water containing 200 ng of total RNA using iScript cDNA Synthesis kit (Bio-Rad Laboratories, Hercules, CA, USA). Real-time PCR was run using a 7500 Real-Time PCR system (Applied Biosystems). Reactions were carried out in triplicate at a final volume of 20 µL containing 20 ng of cDNA template, TaqMan universal PCR Master Mix (2X) and selected TaqMan assays (20X). Samples were maintained at 50 °C for 2 min, then at 95 °C for 10 min followed by 40 amplification cycles at 95 °C for 15 s and at 60 °C for 30 s. The comparative threshold cycle (Ct) method was used to calculate the relative gene expression. The amount of mRNA was normalized to the endogenous genes GAPDH and β-Actin. Reference genes were chosen using the GeNorm VBA applet for Microsoft Excel to determine the most stable ones. Relative quantification of target gene expression was calculated using the comparative Ct method. All experiments were conducted in triplicate.

### 4.9. Senescence-Associated β-Galactosidase (SA β-gal) Staining

The staining was performed following the manufacturer’s instructions (Promega Italia S.r.l., Milan, Italy). Cells were seeded at 20% density in 24 well plates in RPMI 5% charcoal stripped serum. The next day, cells were treated with the above reported concentration of synthetic hormone R1881 or with one of the studied drugs (R-bic or MDV3100). After 24 h cells were irradiated with10 Gy and were incubated for 72 h. They were then washed twice with PBS and fixed for 15 minutes at room temperature. Fixed cells were washed again and incubated overnight with freshly prepared SA β-gal staining solution. Staining solution contains X-gal, a galactopyranoside, which is converted by an active galactosidase into a blue colorant. Blue stained cells were detected and counted by light microscopy. A total of 200 cells/well were counted and the average of triplets was diagrammed.

### 4.10. Western Blot Analysis

Western blot analysis was performed as previously described [48]. Briefly, cells were lysed in RIPA buffer in order to extract total proteins. The same procedure was used in the same manner for all the tested antibodies, according to manufacturer instructions. Mini-PROTEAN TGX™ 4–20% precast gels (BIO-RAD) were run using Mini-PROTEAN Tetra Vertical Electrophoresis Cell and then electroblotted by Trans-Blot Turbo™ Mini PVDF Transfer Packs (BioRad, Milan, Italy). The unoccupied membrane sites were blocked with T-TBS 1X (Tween 0.1%) and 5% non-fat dry milk to prevent nonspecific binding of antibodies and probed with specific primary antibodies (provided in Appendix A) overnight at 4 °C. This was followed by incubation with the respective secondary antibodies. The antibody-antigen complexes were detected by chemiluminescence with Clarity™ Western ECL Substrate (BioRad, Milan, Italy).

### 4.11. Statistical Analysis

Two biological replicates were analyzed for each experiment and the results were reported as mean values and standard deviations (SD). Experimental conditions were compared using Student’s *t*-test for two group comparisons. Differences were considered significant at *p* < 0.05 (*) or *p* < 0.01 (**).

## 5. Conclusions

In summary, our study reveals that AR stimulation by R1881 triggers a finely regulated transcriptional program whose results are visible in SASP activation, followed by DNA repair and cell recovery. In addition, the AR-dependent ATM upregulation is essential for the onset of SASP, as the pharmacological and selective inhibition of ATM prevents the occurrence of the same phenotype, besides rising DNA damage. These results are particularly fascinating because they could represent a key point for the definition of a neoadjuvant ADT in combination with radiotherapy and for the use of e-HFRT. The development of novel therapeutic strategies for prostate cancer is crucial to improve efficacy and tumor cell killing, in particular when considering HFRT which is still barely investigated. This is specifically true for the initial stage of disease, characterized by androgen sensitivity.

## Figures and Tables

**Figure 1 ijms-21-08447-f001:**
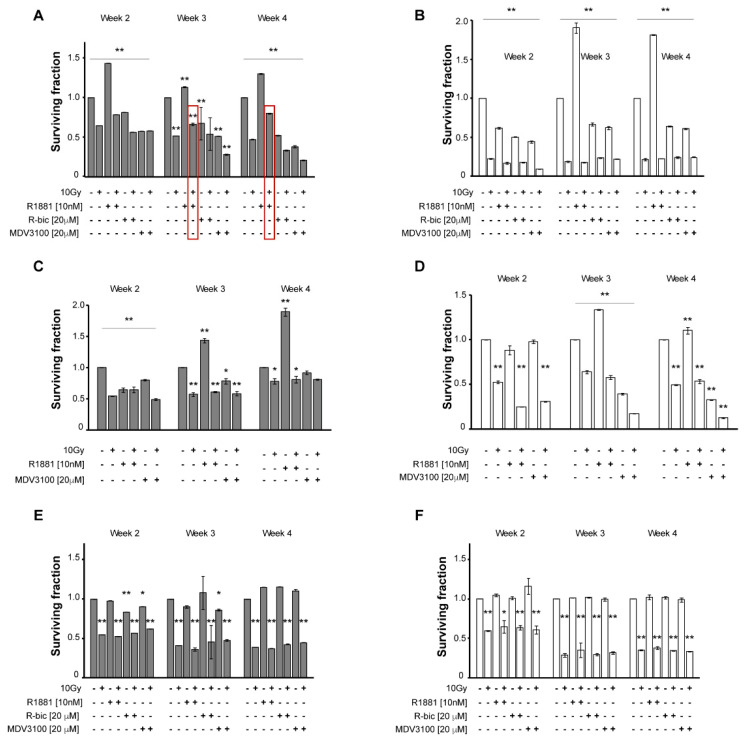
Long term cell viability was explored by soft agar assay. Bar charts show the surviving fraction of cell treatment on the *y*-axis with respect to untreated control cells (1.0 surviving fraction). Hypoxic cell lines are shown on the left and normoxic cell lines on the right. Week 1 was excluded from the analysis because no colonies had formed yet. The mean value and standard deviation of two independent experiments are reported. (**A**,**B**): LNCaP, hypoxic and normoxic conditions; (**C**,**D**): LNCaP R-bic, hypoxic and normoxic conditions; (**E**,**F**): PC3, hypoxic and normoxic conditions. ** *p* < 0.01; * *p* < 0.05 (referred to untreated control cells).

**Figure 2 ijms-21-08447-f002:**
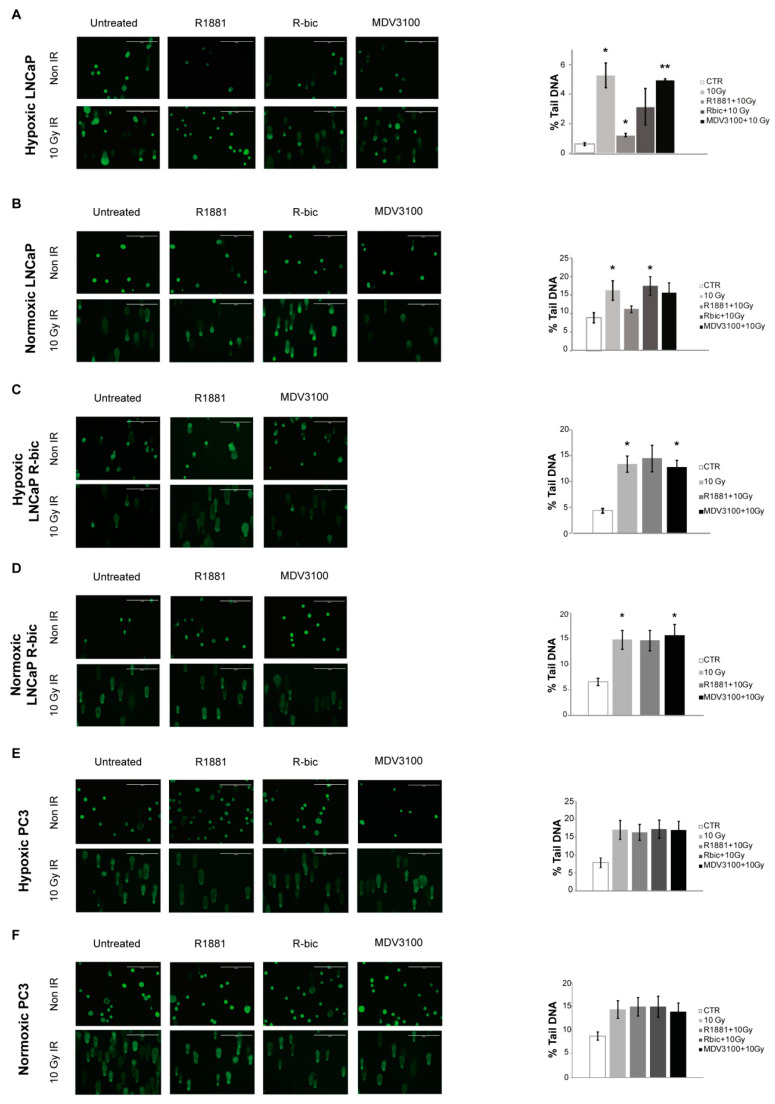
DNA damage assessment in the cell lines studied. Representative images of the different conditions tested are plotted on the left panels of the figure. On the right, bar charts represent the weighted arithmetic mean of tail DNA (calculated by “Mean of % tail DNA for each class” * “%comets of that class/total”) for each irradiated condition. (**A**). LNCaP hypoxia; (**B**). LNCaP normoxia; (**C**). LNCaP R-bic hypoxia; (**D**). LNCaP R-bic normoxia; (**E**). PC3 hypoxia; (**F**). PC3 normoxia. (**) *p* < 0.01; (*) *p* < 0.05. CTR, control; IR, irradiated.

**Figure 3 ijms-21-08447-f003:**
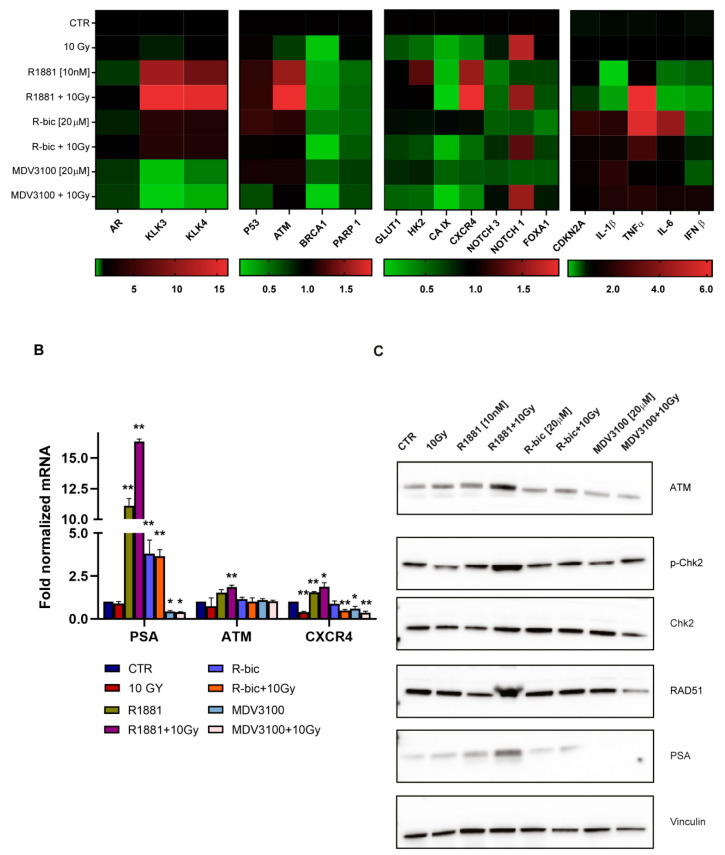
Gene expression profile of hypoxic LNCaP cells for each condition. (**A**). Pathway-clustered heatmap showing the expression patterns of AR (androgen receptor)-, DDR (DNA damage repear)-, HIf1α- and inflammation-related mRNA. Red and green represent up- and downregulated genes, respectively, compared to untreated control cells. (**B**). Real-time PCR of *PSA* (prostate specific antigen), used as positive control for transcriptionally active *AR*, *ATM* and *CXCR4*, with respect to each treatment condition. (**C**). Representative Western blot of hypoxic LNCaP with focus on DDR genes. PSA expression was used as positive control for transcriptionally efficient AR. (**) *p* < 0.01; (*) *p* < 0.05.

**Figure 4 ijms-21-08447-f004:**
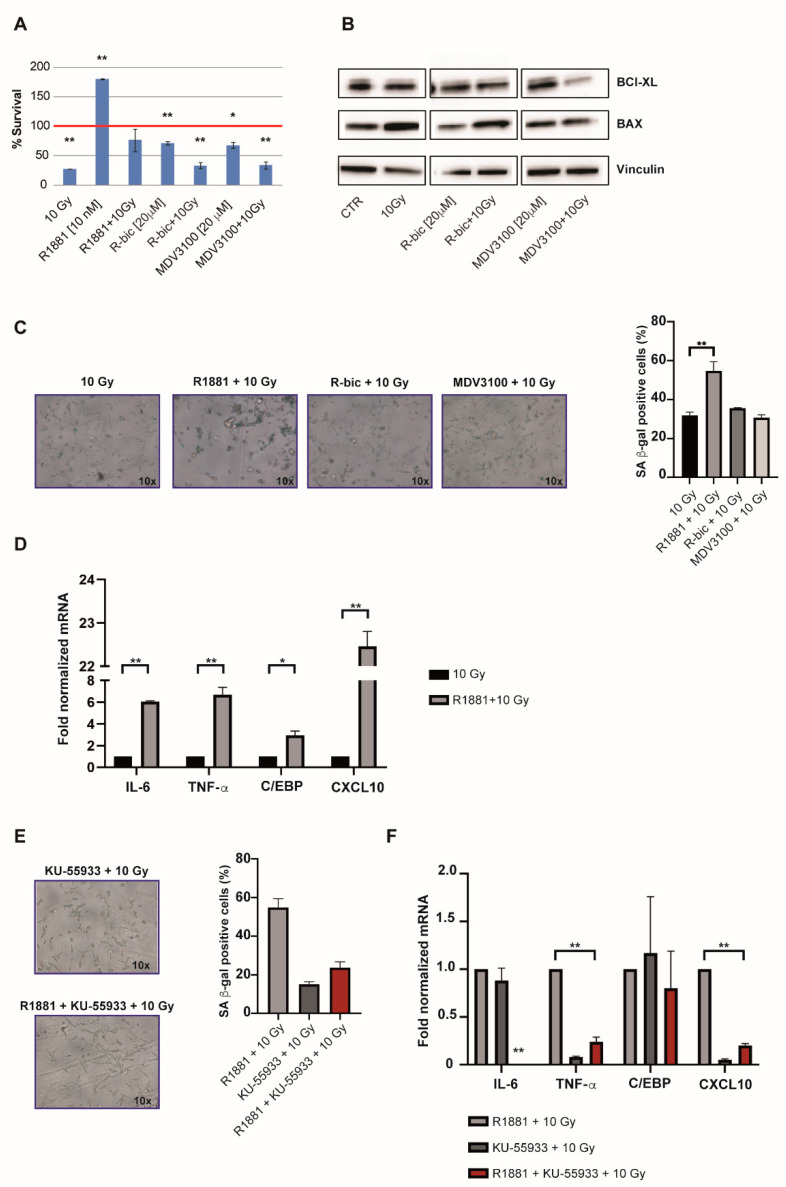
Androgen stimulation by R1881 and 10 Gy induced cell growth arrest and SASP (senescence-associated secretory phenotype). (**A**). Viability analysis of treated cells was investigated by SRB (Sulforhodamine B) assay 144 h after 10 Gy dose. LNCaP were treated with the previously reported concentration of AR modulators for 24 h, prior to irradiation. Subsequently, cells were fixed for 144 h post-IR. Data represent the mean of duplets and each experiment was run in octuplicate. (**B**). Western blot analysis reporting BCl-XL (anti-apoptotic) and BAX (pro-apoptotic) modulation in the “ADT+10Gy” treatment. (**C**,**E**). Cells were fixed and analyzed for SA (senescence-associated) β-Gal activity using a light microscopy. 3x 200 cells were counted and the mean of triplets was diagramed in percentage. (**D**,**F**). Gene expression analysis of IL-6, TNF-α, C/EBP and CXCL10 was performed by qRT-PCR. Gene expression was normalized to ACT B and GAPDH and the values for samples receiving the 10 Gy alone (**D**) and “R1881+10Gy” (**F**) treatment were set as 1. (**) *p* < 0.01; (*) *p* < 0.05.

**Figure 5 ijms-21-08447-f005:**
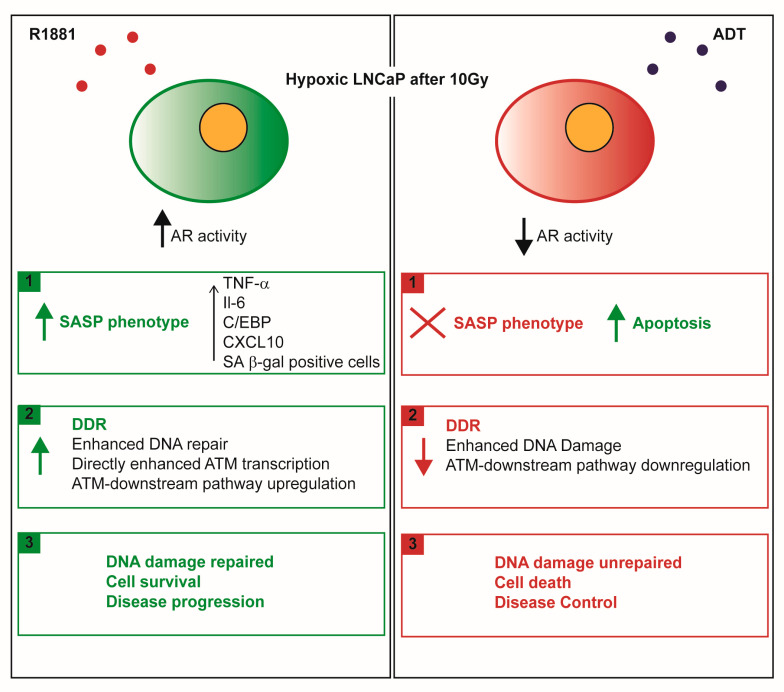
Proposed mechanism of action. After irradiation mimicking an extreme hypofractionated regimen, hypoxic LNCaP, an in vitro model of naïve PCa, takes on a different behavior in presence of androgens or antiandrogens. Using a physiological concentration of androgens (R1881 [10 nM]), AR senses environmental stress caused by the irradiation and triggers the occurrence of SASP, which is responsible for the establishment of a self-protecting program to escape cell death. At the same time, stimulated AR contributes to fueling cancer resistance by giving time to cells to repair DNA damage. This is possible because of the enhanced transcription and transduction of ATM, the cornerstone kinase involved in the DNA damage repair cascade. Conversely, in presence of anti-androgens, e.g., R-bicalutamide or the second generation antiandrogen MDV3100, the same mechanism is not maintained, as confirmed by short- and long-term viability assays and the completely different scenario revealed by gene expression analysis and Western blotting.

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
