# Peer review of "Investigating the Benefit of Combined Androgen Modulation and Hypofractionation in Prostate Cancer"

_ijms, 2020, doi:10.3390/ijms21228447_

Round 1
Reviewer 1 Report
In submitted manuscript Zamagni et al., investigated the therapeutic effects of hypofractionated radiotherapy (HFRT) along with androgen activity modulation in prostate cancer in normoxic and hypoxic conditions, underlying molecular mechanisms that might be involved in shedding lights on the therapeutic outcomes androgen deprivation in combination with radiotherapy in prostate cancer patients.
To this reviewer this manuscript is suitable for publication in IJMS, however some concerns need to be addressed first:
- Authors demonstrated that AR stimulation, in hypoxic conditions, enhanced ATM transcription sustaining malignant SASP and thus supporting cell survival upon HFRT (10Gy). Authors should verify whether the inhibition of ATM restores DNA damage inhibiting SASP in R1881 treated LNCAP cells along with RT.
- Introduction needs to be revised providing more details on the focus of this manuscript.
Minor concerns
Lane 202. Gene expression data are showed in fig. 3D not fig. 4D as stated by authors
Reviewer 2 Report
This was an interesting study. Congratulations on your work!
This manuscript reviews the effects of androgen receptor modulation when combined with hypofractionated radiotherapy in prostate cancer. LNCaP cell lines were compared in normoxic and hypoxic conditions to establish a reference. They then subjected cell lines to 10Gy radiotherapy with and without ADT (bicalutamide or MDV3100) and repeat culture (in addition to an R1881 analysis). The ensuing results of a radiotherapy sensitizing effect in the cell lines even after large doses per fraction radiotherapy is a novel result. They further assess the magnitude of DNA damage in each of the conditions and show both a difference between 10Gy single fraction and 10Gy single fraction + R1881 in hypoxic and normoxic conditions. Finally, the results of their PCR and ATM fluctuation within AR modulated conditions is a significant finding and warrants both validation and further study given the potential radiobiological implications. Overall this manuscript elevates our understanding of the current paradigm in prostate cancer management.
Introduction:
This is appropriately concise and provides a reasonable overview of both the rationale for modest hypofractionation and ADT in prostate cancer and radiobiologic nature of prostate cancer. They may wish to include citation of the appropriate phase III trial evidence; namely CHIP and Profit for modest hypofractionation and hypoRTPC and PACE-B which support extreme hypofractionation.
Results:
The results are well presented and novel considering they present cell survival and hypoxia data after a single high dose of radiotherapy. I would however, suggest the authors in future research consider two fraction regimens such as 24Gy in 2 fractions which may present a different pattern of cell death as seen in clinical practice.
Discussion:
The discussion is appropriately limited to the radiobiological exploration. The authors may wish to add a paragraph to describe potential implications on clinical practice but given the journal audience and lack of a validation cohort this is not necessary.
Materials and Methods:
The methodology is strong and reasonable for this type of study. The only addition I would have is in future studies, the authors should consider using next-generation sequencing or total genomic analyses to determine genomic alterations which one could argue is a new standard.
